# New Insights Regarding Diagnosis and Medication for Schizophrenia Based on Neuronal Synapse–Microglia Interaction

**DOI:** 10.3390/jpm11050371

**Published:** 2021-05-03

**Authors:** Naotaka Izuo, Atsumi Nitta

**Affiliations:** Department of Pharmaceutical Therapy and Neuropharmacology, School of Pharmaceutical Sciences, Graduate School of Pharmaceutical Sciences, University of Toyama, 2630 Sugitani, Toyama 930-0194, Japan; ntk3izuo@pha.u-toyama.ac.jp

**Keywords:** schizophrenia, microglia, synaptic pruning, complement, CX3CR1, medication, diagnosis

## Abstract

Schizophrenia is a common psychiatric disorder that usually develops during adolescence and young adulthood. Since genetic and environmental factors are involved in the disease, the molecular status of the pathology of schizophrenia differs across patients. Recent genetic studies have focused on the association between schizophrenia and the immune system, especially microglia–synapse interactions. Microglia physiologically eliminate unnecessary synapses during the developmental period. The overactivation of synaptic pruning by microglia is involved in the pathology of brain disease. This paper focuses on the synaptic pruning function and its molecular machinery and introduces the hypothesis that excessive synaptic pruning plays a role in the development of schizophrenia. Finally, we suggest a strategy for diagnosis and medication based on modulation of the interaction between microglia and synapses. This review provides updated information on the involvement of the immune system in schizophrenia and proposes novel insights regarding diagnostic and therapeutic strategies for this disease.

## 1. Introduction

Schizophrenia is a severe mental disorder described in the *Diagnostic and Statistical Manual of Mental Disorders*, *Fifth Edition* and the 11th revision of the International Classification of Diseases, and it develops during adolescence and young adulthood in most cases, with a global prevalence of approximately 1% [1]. Patients with schizophrenia mainly exhibit positive symptoms, such as hallucinations and delusions; negative symptoms, such as decreased motivation and anhedonia; and cognitive dysfunction. Previous twin studies on schizophrenia have estimated heritability at 80% [2]. However, the findings from such twin studies may be exaggerated because of environmental and patient lifestyle factors [3]. The significance of the interaction between genetic and environmental factors has been recognized in the pathogenesis of schizophrenia [3].

Genetic studies are effective for determining the causes and pathogenesis of diseases. Genome-wide association studies (GWASs) are important tools for elucidating the mechanisms underlying schizophrenia, including the formation of diseases. The dopamine hypothesis, the current leading theory of the pathogenesis of schizophrenia [4,5], is supported by genetic studies indicating that single nucleotide variations (SNVs) related to dopaminergic transmission, such as DRD2 [6], COMT [7], DISC1 [8], and PCLO [9,10,11], are associated with a higher risk of schizophrenia [5,12]. The GWAS conducted by the Schizophrenia Working Group of the Psychiatric Genomics Consortium has had a notable impact on the study of schizophrenia [13]. The study recruited 36,989 patients with schizophrenia and 113,075 controls and revealed 108 genetic loci associated with the pathology of schizophrenia, including *DRD2* and several genes related to neuronal transmission, which supports the conventional pathological theory of schizophrenia. However, 83 of the 108 loci had not been previously reported when the study was conducted, including genes related to the immune system. It is noteworthy that the major histocompatibility complex (MHC) locus located on chromosome 6 displayed a higher association with schizophrenia than any other locus across the genome, which is consistent with the findings of previous reports [14,15,16,17]. This region contains genes related to innate immunity. These genetic studies have highlighted immune involvement in the etiology of schizophrenia.

The relationship between schizophrenia and the immune system should be investigated. Human postmortem studies of the brains of patients with schizophrenia have previously reported abnormal morphology and the accumulation of microglia, the resident immune cells in the central nervous system [18,19]. In addition, a number of biomarker studies measuring inflammatory cytokines in the peripheral blood and cerebrospinal fluid (CSF) [20,21], and imaging studies using positron emission tomography (PET) to detect neuroinflammation [22,23] have clarified the immunological characteristics of patients with schizophrenia. Immunological changes modify the pathological state of the neuronal system in the brain of patients with schizophrenia [24]. Therefore, the association between immune dysfunction and schizophrenia depends on the background of each individual. A strategy aiming to modulate the immune system could lead to new medications for schizophrenia, and simultaneously, the treatment should be personalized according to certain criteria based on diagnosis via biomarkers to examine the immunological characteristics of each patient.

Dysfunction of the prefrontal cortex (PFC) is considered to be involved in schizophrenia [25,26]. Patients with schizophrenia exhibit volume loss in the PFC [27,28] and disturbance of cognitive functions related to the PFC, including attention, cognitive flexibility, and working memory [29,30,31,32]. For example, in the Wisconsin Card Sorting Test, which assesses the flexibility of thinking associated with the PFC, a higher frequency of perseverative responses has been observed in patients with schizophrenia [31]. Such functional disturbances in the PFC are consistent with the evidence from functional magnetic resonance imaging studies indicating the lower activity of the PFC with respect to episodic encoding and retrieval [33], and in response to consummatory pleasure [34]. In 1982, Feinberg suggested that schizophrenia is caused by a fault in programmed synaptic elimination during adolescence [35]. Thereafter, studies have indicated that synaptic pruning is involved in the development of neuronal circuits, especially in the PFC, in early adulthood, when most individuals are diagnosed with schizophrenia [36]. Furthermore, microglial activations are observed in the PFC of patients with schizophrenia [37]. In a recent decade, synaptic pruning by microglia has been revealed to be involved in psychiatric diseases in addition to normal brain development [38,39,40]. Thus, Feinberg’s suggestion is supported by recent immunological findings in schizophrenia [35].

In the following sections, we introduce microglial functions in synaptic development and maintenance focusing on synaptic pruning, and their relevance to schizophrenia pathology. Afterwards, we provide a novel insight into the diagnosis and treatment of schizophrenia based on microglia–synapse interactions.

## 2. Microglia–Synapse Interaction

Microglia are brain cells related to innate immunity, and they are involved in various physiological and pathological processes [41]. Similar to peripheral macrophages, the activation of toll-like receptors on the microglial surface mediates their morphological changes and induces phagocytosis and the release of inflammatory cytokines [42,43]. Microglia are activated in disease-specific and aging-specific manners to release inflammatory cytokines and chemokines, and they remove the deposits of pathological proteins by the process of phagocytosis [44]. In recent decades, it has been revealed that the physiological functions of microglia are necessary to construct and maintain healthy neuronal circuits. Microglia dynamically extend their processes to neuronal synapses to survey their conditions [45,46,47,48]. The microglia eliminate unnecessary synapses (a process called “synaptic pruning”) during the appropriate and specific periods of neuronal development [38,39]. Neuronal circuits then become sophisticated through the selective pruning of synapses with lower activity [39]. The pharmacological depletion of microglia by antagonism of colony-stimulating factor 1 receptor (CSF1R) from postnatal day 2 (P2) to P13 in mice disrupted synaptic pruning and induced poly-innervation of the medial nucleus of the trapezoid body neurons, instead of healthy mono-innervation [49]. CSF1R antagonism from P14 to P28 increased synaptic density due to decreased pruning and disturbed glutamatergic transmission [50]. These findings clearly demonstrated the significance of synaptic pruning by microglia in neuronal development. 

Synaptic pruning and phagocytosis share a common machinery mainly regulated by “find-me”, “eat-me”, and “don’t-eat-me” signals [51,52]. The find-me signal is mediated by chemo-attractants and their receptors. Chemokine (C-X3-C motif) ligand 1 (CX3CL1), which is highly expressed in the brain especially in the cerebral cortex and hippocampus [53], is secreted from neurons or synapses to bind to its receptor CX3C chemokine receptor 1 (CX3CR1) and is exclusively expressed in microglia [54]. Each line of *Cx3cl1* and *Cx3cr1* knockout mice exhibited delayed and reduced synaptic pruning during the developmental period [38,39]. The secretion of mature CX3CL1 requires cleavage by a disintegrin and metalloproteinase domain-containing protein 10 (ADAM10) on the plasma membrane [55]. The pharmacological inhibition of ADAM10 further leads to profound deficits in the elimination of sensory synapses [55]. ATP is extracellularly released through the pannexin-1 channel to bind purinergic receptors [56]. P2Y12 mediates microglial recruitment to remove apoptotic cells as well as unnecessary synapses [57,58,59,60]. Activity-dependent synaptic pruning during development was shown to be delayed in P2Y12-deficient mice [58,59,60].

Phosphatidylserine, usually existing on the inner leaflet of the plasma membrane, mediates eat-me signals when exposed on the cellular surface [61,62]. Phospholipid scramblase 1 [62] and transmembrane protein 16F (TMEM16F) [63], which are calcium-activated phosphatidylserine scramblases, reversibly expose phosphatidylserine on the cell surface. Although these scramblases are expressed by neurons and modulate microglia [62], their functions in synaptic pruning are unclear. Among various receptors of phosphatidylserine, triggering receptors expressed on myeloid cells 2 (TREM2) and G protein-coupled receptor 56 modulate microglia to induce synaptic pruning and phagocytosis [64,65].

The complement system, discovered by Bordet and Gengou in 1901, is composed of nine main components (C1–C9), some inhibitory and regulatory substances, and their membrane receptors (Figure 1) [66,67,68]. They exert their functions, such as the opsonization of antigens, the formation of the membrane attack complex (MAC) to destroy bacteria, and the stimulation of macrophage chemotaxis, via three pathways of enzymatic chain reaction (classical, alternative, and lectin pathways). In addition to the long-studied, immune-activating effects, novel functions of complements of synaptic pruning during development have been revealed [69,70]. C3, expressed in the synaptic regions as a “tag”, which allows synapse elimination, mediates synaptic pruning and regulates synaptic density and transmission via its receptor, CR3, on microglia [71,72]. Synaptic C1q and C4 exert a similar tagging function for synaptic pruning [73,74]. The proteomic analysis of the individual synaptosomes conducted using flowcytometry revealed that the local expression of proteins related to neuronal transmission, energy metabolism, and the antioxidant system was altered in the C1q-tagged synaptic fraction [73]. Although it is unclear whether such an alteration is a trigger or consequence of C1q tagging, these proteins altered in C1q-tagged synaptic fractions possibly change the neuronal transmission of synapses.

Cell-surface sugar residues further regulate microglial phagocytosis. Galactose promotes phagocytosis by binding to the galectin family [75], and sialic acid suppresses this function via Siglec [76,77]. The heterozygous mice with glucosamine-2-epimerase/N-acetylmannosamine kinase, a sialic acid synthase, exhibited a decrease in postsynaptic marker PSD-95 accompanied by a decrease in sialic acid in the brain and altered the microglial morphology [78]. The don’t-eat-me signal, similar to sialic acid, is mediated by CD47 and its receptor signal-regulatory protein alpha (SIRPα). CD47 suppresses synaptic elimination and phagocytosis by microglia [79].

## 3. Microglia–Synapse Interaction in Schizophrenia

Although synaptic pruning by microglia is essential for the sophistication of neuronal circuits during the developmental stage, excessive elimination is considered to be involved in neurological disorders (Figure 2). As predicted by Feinberg, synaptic pruning might be involved in the pathogenesis of schizophrenia [35], and studies have suggested the presence of excessive synaptic pruning in schizophrenia. Recently, Sekar et al. revealed that an allele of C4 located in the MHC locus increases the risk of schizophrenia [74], which has led to studies searching for a link between the complement system and schizophrenia [80]. The association between C4 and schizophrenia is supported by a study from Sweden that proposed the possibility of predicting the future risk of this disorder by utilizing blood analysis during the neonatal period [81]. Higher C4 levels were likewise detected in the CSF of patients with schizophrenia [82]. The SNVs of the C3 gene with protective or harmful effects regarding schizophrenia onset were identified by a study that recruited more than 2000 Han Chinese individuals [83]. C3 mRNA expression was upregulated in the PFC of those that demonstrated depression followed by suicide [84]. Moreover, the C5 protein levels in the CSF were elevated in patients with schizophrenia [85].

Since alterations in the complement system of the brain are associated with psychiatric disorders [74,81], identifying their pathological contribution should help clarify the mechanisms underlying these diseases. Mice overexpressing C4 were generated to investigate the pathological significance of C4 upregulation in schizophrenia [86]; the overexpression of C4 in utero in the PFC of the mouse resulted in a reduction in spine density due to abnormal increase in synaptic removal by microglia 21 days after birth and a related decrease in the frequency and amplitude of miniature excitatory postsynaptic potentials [86]. These abnormal neuronal connections caused a decrease in social interaction with the mother at the age of 60 days [86]. Yilmaz et al. generated mice overexpressing human C4A with a knockout of the murine C4 gene [87]. These mice exhibited enhanced microglial synaptic pruning in the PFC and a reduction in spine density at 60 days of age [87]. In addition to decreased social behavior, these mice showed increased anxiety-like behavior and decreased novel environment exploration [87]. Such behavioral alterations observed in both studies are assumed to be equivalent to a part of a schizophrenia symptom. These studies strongly suggest a link between elevated C4 expression associated with this disorder and the pathogenesis of schizophrenia, including spine density reduction in the PFC.

The complement-related synaptic impairment in schizophrenia was revealed by a study with patient-derived induced pluripotent stem cells (iPSCs) [88]. Interestingly, excessive synaptic pruning observed in schizophrenia appears to involve the impairments of both neurons and microglia. The microglia-like cells (iMG) differentiated from the monocytes of patients with schizophrenia exhibited higher phagocytic activity compared to that of synaptosomes prepared from neurons derived from iPSCs of healthy controls and of patients with schizophrenia. Although there were no differences in spine density between the neuronal cultures derived from iPSCs of controls and of patients with schizophrenia, C4 risk variants increased C3 deposition in the latter culture. Furthermore, the co-culture of neurons derived from the iPSCs of patients with schizophrenia with the corresponding iMG showed a reduction in spine density compared to what was prepared from healthy subjects. These results support the notion of excessive synaptic pruning mediated by complements in schizophrenia [88]. The results are expected from the mechanistic analysis on the malignancy of neurons and microglia in schizophrenia.

The deregulation of brain complements, which is closely related to psychiatric disorders including schizophrenia, is derived from genetic and/or environmental factors. Maternal infection during the perinatal period is known to increase the risk of schizophrenia [89,90]. Based on the evidence that maternal infection reduces synaptic density in the brain of the offspring [91,92], such an infection has certain impacts on the brain complement system and possibly increases the risk of schizophrenia. Higher levels of maternal blood C1q have been reported to be associated with the risk of schizophrenia [93]. In this report, adenovirus, herpes simplex virus 2, influenza B virus, and *Toxoplasma gondii* approximately doubled the risk. There was no correlation between C-reactive protein levels and schizophrenia risk [93]. *Toxoplasma* infection in humans is a risk factor for schizophrenia, and it induces behavioral dysfunction in rodent models [94,95]. Severe infection of adult mice with *Toxoplasma* has been reported to greatly upregulate C3 and C4 proteins, and mildly enhance C1q expression [96,97]. In rodent studies, the induction of marked systemic inflammation by lipopolysaccharide administration increased the levels of C3 and C3R in the brain [98,99].

Synaptic pruning is further regulated by signaling other than the complement factors. PolySia is a linear polymer of sialic acid with a degree of polymerization of 8–400, mediating a don’t-eat-me signal to microglia [100]. The number of hippocampal and dorsolateral PFC cells with polySia is reduced in the brains of patients with schizophrenia [101,102]. The content of the polisialylated neural cell adhesion molecule (PSA-NCAM) is further reduced in the PFC of patients with schizophrenia [102]. The serum PSA-NCAM is associated with cognitive decline, as evaluated by the Mini-Mental State Examination, and is further associated with the reduction in gray matter volume [103]. Furthermore, the SNVs on *ST8SIA2*, encoding one of the polysialyltransferases, are related to schizophrenia. Associations between rs3759916 and rs3759914 in the promoter regions of *ST8SIA2* and schizophrenia have been identified in the Japanese population [104]. Another association between rs3759915 and schizophrenia has been found in Chinese [105] and Spanish populations [106]. In rodent experiments, *ST8SIA2* knockout mice exhibited disturbed mossy fiber formation and impaired fear-conditioned memory [107]. TREM2, a phosphatidylserine receptor mediating eat-me signaling, is important for adequate neuronal development by modulating microglia [64]. TREM2 mRNA levels are increased in the peripheral leukocytes of patients with schizophrenia [108,109,110], which is consistent with the results of rodent studies. However, the pathological roles of these sugar residues, phospholipids, and related molecules are not fully understood. Further studies clarifying such molecules will lead to the precise understanding of microglial synaptic pruning in schizophrenia.

## 4. Potential Diagnosis/Medication for Schizophrenia Based on Microglia–Synapse Interaction

### 4.1. Perspective for the Diagnosis of Schizophrenia

Although the deficits in synaptic elimination are involved in the pathogenesis of schizophrenia, their contribution will differ among patients because these deficits are derived from the genetic and/or environmental background of the individuals. Obtaining information on patient’s brain status is expected to help determine the therapeutic direction for patients with schizophrenia. PET imaging, which visualizes the distribution and behavior of a specific molecule, is generally utilized. Based on the notion that excessive synaptic elimination mediates the pathology of schizophrenia, monitoring the synaptic density or pruning capacity of microglia would help in diagnosis. A small-scale PET study by Howes et al. reported a reduced synaptic density in the PFC of patients with schizophrenia, caused by the reduced binding of the radioactive ligand [^11^C]UCB-J to its target protein, synaptic vesicle glycoprotein 2A [111,112]. Radio-tracing is strongly expected to be used in the context of schizophrenia to enhance clinical evaluation. With respect to the microglial proteins monitored by PET tracers, the translocator protein 18 kDa (TSPO) has been targeted in neurological diseases, such as Alzheimer’s disease and multiple sclerosis [113]. However, TSPO tracers were reported to be unable to discriminate patients with schizophrenia from healthy controls [114]. This is possibly because TSPO is upregulated in the pro-inflammatory context in glial cells [115], while the morphology of microglia with longer processes and more branches observed in schizophrenia are different from those in Alzheimer’s disease and multiple sclerosis [18,19]. In order to discriminate patients with schizophrenia, a target that reflects microglial pruning activity is suitable. Scarce radiotracers targeting anti-inflammatory microglia have been developed [116]. Recently, a radioactive ligand that binds to the purinergic receptor P2Y12 was generated and appears to be undergoing structural optimization to enhance blood–brain barrier (BBB) penetration [116]. CX3CR1 and complement receptor CR3 are candidate targets on the cell surface, reflecting phagocytosis and synaptic pruning activity. AZD8797 and SB290157 are selective antagonists of CX3CR1 and CR3 [117,118], respectively, and have the potential to be the lead compounds utilized as radioactive tracers for PET imaging to monitor the synaptic pruning activity of microglia.

The blood biomarkers that reflect the molecular condition of the brain are highly desirable because proteins derived from the brain are mixed with those from peripheral tissues in the circulation and are difficult to discriminate. In schizophrenia, the blood complement levels in patients are not notably altered. In one report, there were no differences in serum C1q, C3, and C4 in patients with first-episode psychosis (FEP) occurring at approximately 20 years of age, but a 20% increase in C4 levels was noted in patients more than seven years after onset [119]. Another study reported a 20% increase in the blood levels of C4 and C9 in drug-free FEP [81]. A meta-analysis of blood complements in patients with schizophrenia revealed no differences compared to the controls [120]. Elevated C4 levels have simultaneously been found in the CSF of patients with schizophrenia [82], and another report further found elevations of C5 in the CSF of patients with this disorder [85], while the blood leukocyte fraction from drug-free FEP showed no change in C4 mRNA [121]. Therefore, complement upregulation specifically occurs in the brain.

The exosome, which is a nano-sized carrier with a lipid bilayer and is secreted into the extracellular space, is attracting attention as a promising tool for disease diagnosis [122,123]. Cells actively enclose proteins and nucleic acids into exosomes to transmit signals to neighboring or distant cells [122,123]. In recent decades, a technique has been developed for extracting cell-type specific exosomes from blood samples via surface markers. Goetzl et al. developed a method to collect neuron-derived exosomes (NDE) using the neuronal surface marker L1CAM [124,125]. Since the proteins and nucleic acids included in the exosome could escape from nonspecific enzymatic degradation in the blood, their methods for collecting NDE made it possible to understand the molecular circumstances in the patient’s brain. This method has already been applied to some neurological diseases, including schizophrenia. Goetzl et al. further found a severe reduction in MFN2 and CYPD proteins in the NDE of patients with schizophrenia, suggesting the impairment of mitochondria in the disease [126]. They collected astrocyte-derived exosomes (ADE) to determine the upregulation of complement proteins; this suggests that complement abnormalities occur in astrocytes in schizophrenia [127]. Currently, there are limited reports on the collection of microglia-derived exosomes (MDE) [128] because microglia largely share the surface marker with peripheral macrophages. Thus, targeting microglial-specific surface markers, such as TMEM119 [129], could be beneficial for collecting MDE. Since the complement system functions through cellular interaction, the application of a combination of NDE, ADE, and MDE would be a powerful tool to monitor complement abnormalities in the brain.

### 4.2. Perspective for Medication Drug Therapy in Schizophrenia

Clinical trials of anti-inflammatory drugs for schizophrenia have been performed. Minocycline, a tetracycline-type antibiotic, suppresses the microglial inflammatory response [130,131]. Minocycline has been reported to rescue cognitive dysfunction and social behavioral impairment in animal models of schizophrenia [132,133]. Minocycline abolished the phagocytosis of patient-derived iMG toward spine density on neurons differentiated by patient-derived iPSCs [88]. In clinical trials, minocycline treatment improved the working memory [134], and verbal and visual learning [135] of patients with schizophrenia. Although minocycline is infrequently prescribed partly because of the increased risk of autoimmune disease [136], these outcomes demonstrate the clinical effectiveness of this strategy targeting microglia in schizophrenia. N-acetylcysteine (NAC), a precursor of antioxidant glutathione, exerts a wide range of protective effects, such as the regulation of oxidative status, inflammation, and monoamine neurotransmission in rodent models and human patients [137,138]. NAC showed a beneficial effect mainly on the negative symptoms of patients with schizophrenia receiving antipsychotic treatment (comprehensively reviewed in [139]). It is possible to achieve these therapeutic effects of NAC by multiple mechanisms, such as modulation of the oxidative and inflammation statuses, and neurotransmission.

The complement components are considered not only as diagnostic biomarkers but also as therapeutic targets. This is partly because the overexpression of C4 in mice induces behavioral impairments related to schizophrenia, which is accompanied by excessive synaptic pruning by microglia and defects in neuronal transmission in the PFC [86,87]. In addition, C3, C4, and C5 are considered to be upregulated in the brains of patients with schizophrenia [81,82,83,84,85]. In 2007, eculizumab, a monoclonal antibody against C5, was approved as a drug for paroxysmal nocturnal hemoglobinuria (PNH) by the FDA [140,141]. However, eculizumab stochastically increases C3 expression in erythrocytes to mediate the formation of MAC leading to de novo extravascular hemolysis in patients with PNH, which reduced the clinical benefits of the treatment [142]. To overcome this side effect, pegcetacoplan, a peptide-based C3 inhibitor, was developed and administered in an open-label, phase Ib, prospective, and non-randomized study [143]. Pegcetacoplan was confirmed to be well-tolerated and resulted in an improved hematological response in patients with PNH who remained anemic during treatment with eculizumab [143]. Another potential modulator of synaptic pruning for schizophrenia treatment targets CX3CL1-CX3CR1 signaling. E6011, a monoclonal antibody that selectively binds to CX3CL1, has been developed for rheumatic diseases [144]. Recently, a phase 2, multicenter, randomized, double-blind, and placebo-controlled study of E6011 for rheumatoid arthritis was conducted after recruiting 190 patients with active rheumatic arthritis (RA) who inadequately responded to methotrexate, the first-choice drug for RA treatment. The administration of E6011 once every two weeks for 24 weeks significantly improved the clinical RA score, which was evaluated using the American College Rheumatology 20% improvement criteria without severe adverse events [145,146]. A clinical trial of E6011 for Crohn’s disease has likewise started [147]. Although clinical studies have not been initiated for schizophrenia, modulators of the C3-CR3 and CX3CL1-CX3CR1 pathways could be potential candidates for future medications for the behavioral and cognitive symptoms of schizophrenia mediated by decelerating excessive synaptic pruning.

The activation of the don’t-eat-me signal should be effective for schizophrenia medication. The stimulation of this signal by CD47-SIRPα mediates escape from synaptic pruning [148,149]. In preclinical studies, the deficiency of CD47 induced impairment of social behavior [150] and exaggerated cognitive dysfunction in mice [151]. The agonism of SIRPα could be a potential strategy to suppress synaptic pathology in schizophrenia.

Considering that excessive synaptic pruning occurs before the emergence of clinical symptoms in schizophrenia, the strategy to modulate synaptic pruning should start preclinically. Thus, such medication should be combined with the monitoring of the synaptic status in the brain as mentioned above. In addition, there is an increased risk of infection because of the suppression of innate immunity, which could ensue with these medication candidates.

It is necessary for drugs to reach the brain for the treatment of schizophrenia. With respect to recent drug delivery technology into the brain, conjugations by BBB-penetrating peptides [152,153] and intranasal administration have been developed [154,155]. The combination of the synapse-protecting reagent and brain delivery methods could be used to treat schizophrenia.

## 5. Conclusions

Microglia have been suggested to play a significant role in excessive synaptic elimination with respect to the pathology of schizophrenia. Various molecules, such as CX3CL1-CX3CR1, CD47-SIRPα, and lectins, are physiologically and pathologically involved in this process. Recent publications indicating the contribution of complement components to schizophrenia are attracting attention with respect to this relationship. The monitoring and modulation of microglial synaptic pruning based on the complement system would be powerful tools for diagnosis and medication in schizophrenia. Further investigation on synapse–microglia interaction could reveal other molecular targets for diagnosis and medication. Considering that schizophrenia is a polygenic disease, such medication and diagnosis should be combined to find excessive pruning in the preclinical stage, and early medication could maximize the therapeutic effects for schizophrenia. Developments of the medication and diagnosis focusing on synapse–microglia interaction and its clinical application to schizophrenia are desired.

## Figures and Tables

**Figure 1 jpm-11-00371-f001:**
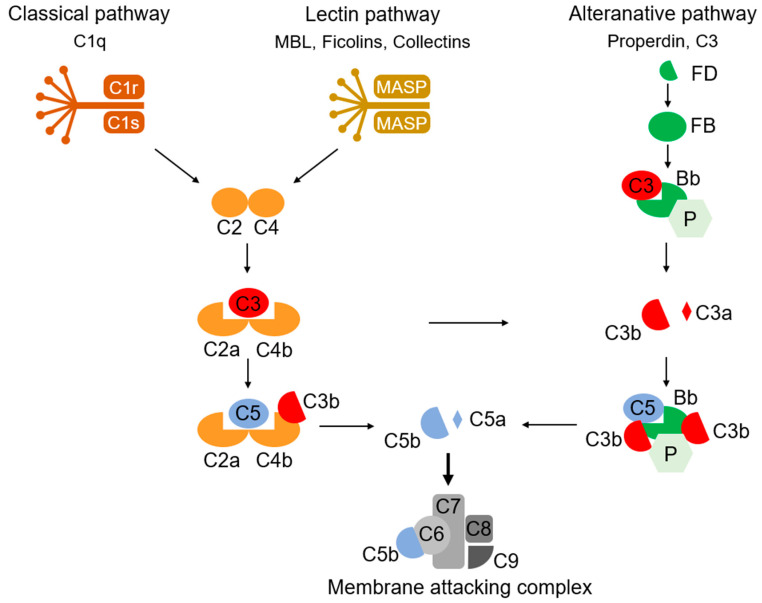
An overview of systemic complement activation. The complement system is composed of nine main components (C1–C9) and regulatory factors. Immunological functions are conducted by three pathways: the classical, alternative, and lectin pathways. In these pathways, enzymatic chain reactions of the complements proceed to finally form a membrane attack complex to destroy bacteria and virus-infected cells. MASP, mannose-binding protein-associated serine protease; FD, factor D; FB, factor B; Bb, factor Bb; P, properdin.

**Figure 2 jpm-11-00371-f002:**
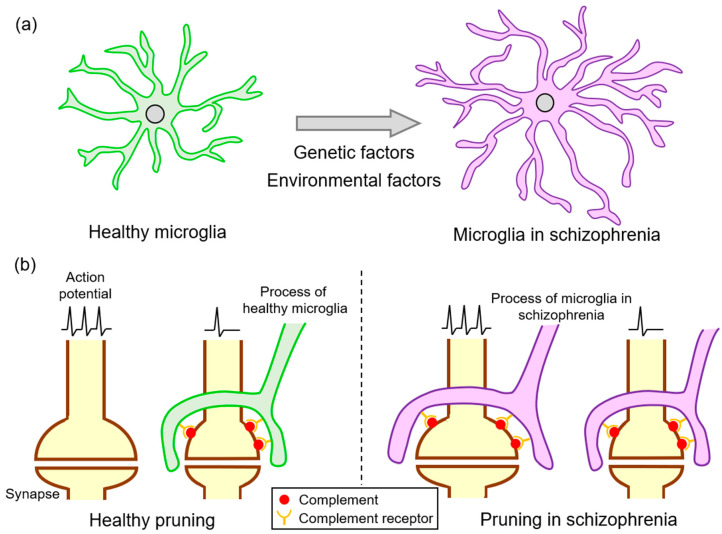
The microglial synaptic pruning in schizophrenia. (**a**) Microglia exhibit hyper-ramified morphology in schizophrenia due to genetic and environmental factors. (**b**) Healthy microglia eliminate complement-tagged synapses with lower activity. In schizophrenia, hyper-ramified microglia prune healthy synapses by the excessive activation of complements.

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
