# Peer review of "New Insights Regarding Diagnosis and Medication for Schizophrenia Based on Neuronal Synapse–Microglia Interaction"

_jpm, 2021, doi:10.3390/jpm11050371_

Round 1
Reviewer 1 Report
I feel that this narrative review is well written with a thorough review of the relevant literature and nice conclusions. However, I think you should mention Sellgren et al 2019 Nature Neuroscience(PMID 30718903). This paper investigates the role of C4 on synaptic pruning in an IPS cell-based model.
Author Response
Reply to Editor’s comments
Reviewer 1
> I feel that this narrative review is well written with a thorough review of the relevant literature and nice conclusions. However, I think you should mention Sellgren et al 2019 Nature Neuroscience (PMID 30718903). This paper investigates the role of C4 on synaptic pruning in an IPS cell-based model.
Response:
We appreciate your supportive comment and suggestion to cite the important study: “Increased synapse elimination by microglia in schizophrenia patient-derived models of synaptic pruning” by Sellgren et al that was published in Nature Neuroscience in 2019. This report indicates the involvement of the complement system in “patients” with schizophrenia, which strongly supports the conclusions of our review. We have cited the suggested reference as [88] and have introduced it in lines 181–192 of the revised manuscript as follows.
- Sellgren, C. M.; Gracias, J.; Watmuff, B.; Biag, J. D.; Thanos, J. M.; Whittredge, P. B.; Fu, T.; Worringer, K.; Brown, H. E.; Wang, J.; et al. Increased synapse elimination by microglia in schizophrenia patient-derived models of synaptic pruning. Nat. Neurosci. 2019, 22, 374-385, doi: 10.1038/s41593-018-0334-7.
Complement-related synaptic impairment in schizophrenia was revealed by a study with patient-derived induced pluripotent stem cells (iPSCs) [88]. Interestingly, excessive synaptic pruning observed in schizophrenia appears to involve the impairments of both neurons and microglia. Microglia-like cells (iMG) differentiated from the monocytes of patients with schizophrenia exhibited higher phagocytic activity to that of synaptosomes prepared from neurons derived from iPSCs of healthy controls and of patients with schizophrenia. Although there was no difference in spine density between the neuronal cultures derived from iPSCs of controls and of patients with schizophrenia, C4 risk variants increased C3 deposition in the latter culture. Furthermore, co-culture of neurons derived from iPSCs of patients with schizophrenia with the corresponding iMG showed reduction of spine density compared to that prepared from healthy subjects. These results support the notion of excessive synaptic pruning mediated by complements in schizophrenia [88]. It is expected that mechanistic analysis on the malignancy of neurons and microglia in schizophrenia.

Reviewer 2 Report
New insights regarding diagnosis and medication for schizophreniabased on neuronal synapse-microglia interaction
Authors: Nitta, A., Izuo, N.
In this manuscript, the authors discuss the role of excessive microglialsynaptic pruningin the development of schizophrenia (SCZ). The authors have made an effort to evaluateinsights regarding diagnosticandtherapeutic strategiesfor SCZ.Given the interest in new insights for medication to improve cognitive functioning in SCZ, this manuscript makes an important and interesting contribution to the literature. Yet, the manuscript lacks some thoroughness in writing up details, connecting main concepts with each other, correct grammar, proper usage of the English language, etc.Some notes for improvement:
General comments:
•Overall writing:the writing styleand appropriate usage of the English languagecan be improved to obtain a higher quality of academic writingand to enablebetterreadability of the manuscript.For example, line 55 –59: “So far, genetic analyses including GWASs revealedschizophrenia-associated sequencevariations related to dopamine transmission, such as DRD2 [7], COMT encoding catechol-O-methyltransferase [8], DISC1 encoding disrupted in schizophrenia 1 [9], and PCLO encoding piccolo [10,11,12], whose single nucleotide variations (SNVs) that increase the risk of this disorder have been identified[reviewed in 5 and 13].”or line 104 –106: “In the developmental stage, microglia promote synaptogenesis and eliminate unnecessary synapses (a process called “synaptic pruning”) during different periods of circuit development[37, 38]”. In addition, when using abbreviation, the authors should make sure to write them out at least once in the manuscript (e.g., mEPSP).
•While the authors extensively refer to other studies, they rarely provide details on the applied methods or exact results, i.e., findings from other studies are often formulated in a way that leaves them ambiguous which increases the risk of missing important information. Some examples:
o Line 83 –84: “Cognitive functions related to PFC, including attention, cognitive flexibility, and working memory, are disturbed [29-32], which has been confirmed by functional magnetic resonance imaging [33, 34].” –specify in what way cognitive functions are disturbedand how it has been confirmed by fMRI. Additionally, neither ref. [33] nor ref. [34] made claims about the cognitive functions mentioned by the author –this illustrates the importance on providing details onthe exact results.
o Line 144 –148: “Proteomic analysis of the synaptic microregion revealed that the local expression of proteins related to neuronal transmission, energy metabolism, and the antioxidant system was altered in the C1q-tagged synaptic fraction [70]. Although it is unclear whether such an alteration is a trigger or a consequence of the tagging of C1q, the properties of tagged synapses were demonstrated to differ from those of untagged synapses.” –specify in what way the local expression of these proteins is altered and how the properties of tagged and untagged synapses differ from each other. Additionally, ref. [70] found that out of 410 spots, only 25 exhibited significant differences between tagged and untagged samples, which is not a particularly strong result –this illustrates the importance on providing details on the applied methods.
o Line 284 –285: “Recently, a phase 2, multicenter, randomized, double-blind, placebo-controlled study offered a positive result.” –specify what this ‘positive result’ entailed.
•The main concepts introduced in this manuscript are not clearly linkedto each other and often appear to be a lose enumeration of information. For example: “Psychological stress also increases the risk of SCZ” –the added value of this information is unclear given that this concept is not mentioned again in the manuscript/has not been linked to other concepts such as complement or microglia. It is strongly suggested to provide a clear link between the individual concepts. See,for instance,Germann, M., Brederoo, S. G., & Sommer, I. E. (2021). Abnormal synaptic pruning duringadolescence underlying the development of psychotic disorders.Current opinion in psychiatry.for a nice example on how to link complement, microglia, stress, astrocytes and cognitive deficits in SCZ.
•Although the reference list of this manuscript is quite extensive, more than 60% references are from 2016 or older. Taking into account the main aim of this manuscript to “offer recentknowledge of SCZ [...]”, it is suggestedto substitute older references with more recent literatureto meet the aim of the manuscript.For example:
o Reference [1] on the global prevalence of SCZ stems from 2005.
o The presented view on the microglial pro-inflammatory phenotype is outdated(see for instance Dubbelaar ML, KrachtL, Eggen BJL, Boddeke EWGM. The kaleidoscope of microglial phenotypes. Front Immunol 2018; 9:1753. doi: 10.3389/fimmu.2018.01753.).
•This manuscript could benefit from summarizing BethStevensline of research on complement mediated microglial synaptic pruning.
Introduction:
•This manuscript clearly supports the excessive pruning hypothesis on SCZas proposed by Feinberg.In the introduction, however, the dopamine hypothesis on SCZ is discussed. The added value of this discussion is unclear, especially given that this topic does not come back later on in the manuscript. Given themain topic of the manuscript “synapse-microglia interaction” and the hypothesisstating thatexcessive synaptic pruning mediated by microglial overactivation play a role in the development of SCZ,it would bemore relevant to mention that current antipsychotic medication is not effective in treating cognitive symptoms(after all, excessive pruning is expected to underlie the cognitive functioningdeficitsin SCZ).
•The conclusions drawn by the authors at the end of the introduction (i.e., “Thus, Feinberg’s suggestion is supported by recent immunological findings in schizophrenia.”) is not properly backed up; the paragraph does not contain a single immunologicalfindingto base this conclusion on.
•The statements made by the authors in lines 75 –77 are not backed up properly withreferences.
Potential diagnosis/medication for schizophrenia based on microglia-synapse interaction:
•The authors provide a well-structured and cleardescriptionofperspectivesfor diagnosis ofSCZ. However, the section on perspective for medication drug therapy in SCZneeds to be improved: oThe authorsmention that eculizumab, a drug for paroxysmal nocturnal hemoglobinuria (PNH),has been approvedby the FDA. Isthere any literature on the exact effectof eculizumabin SCZ? What is the (expected)beneficial effect on behavioral/cognitive functioning in SCZ?oTheauthorsfurtherdiscussE6011which has been developed for rheumatic diseases and mention thata phase 2, multicenter, randomized, double-blind, placebo-controlledstudy showed a positive result.What is this ‘positive result’(beneficial to mention effect sizesor similar)? In which population(s)was this drug tested?What is the (expected) beneficial effect on behavioral/cognitive functioning in SCZ?
o Theauthors proposed CD47asastimulant for the “don’t-eat-me” signalwhich is predicted to decrease synaptic pruning.Are there any studieswhichinvestigatedthis drug(possibly in SCZ)?If so, what were the outcome(s)?Furthermore, if the assumption is thatexcessive synaptic pruning in SCZ occurs during adolescence/young adulthood, this drug is ratherutilized aspreventivetool, not as a treatment for SCZ.
o Both minocycline and N-acetyl-cysteine (NAC) have been demonstrated to be effective, safe, and well-tolerated in SCZwith few side-effects. Why havethese drugs not been mentioned?
Conclusion:
•This manuscript aims to provide new insights regarding diagnosis and medication for SCZ based on neuronal synapse-microglia interaction.The conclusion, however, is formulated in a way that leaves insights for diagnosis and medication for SCZ ambiguousand vague.
•In the introduction of this manuscript,it is stated that “In addition, a number of biomarker studies measuring inflammatory cytokines in the peripheral blood and cerebrospinal fluid (CSF) [21, 22] and imaging studies using positron emission tomography (PET) to detect neuroinflammation [23, 24]have clarified the immunological propertiesof patients with schizophrenia.”–this is in conflict with the last sentence of the conclusion statingthat an immunological understanding has not yet been reached.
Author Response
Reviewer 2
> In this manuscript, the authors discuss the role of excessive microglial synaptic pruning in the development of schizophrenia (SCZ). The authors have made an effort to evaluate insights regarding diagnostic and therapeutic strategies for SCZ. Given the interest in new insights for medication to improve cognitive functioning in SCZ, this manuscript makes an important and interesting contribution to the literature. Yet, the manuscript lacks some thoroughness in writing up details, connecting main concepts with each other, correct grammar, proper usage of the English language, etc. Some notes for improvement:
Response:
We appreciate your constructive and helpful comments. We have responded to your comments in a point-by-point manner below and have revised the manuscript accordingly. Moreover, our whole manuscript has been re-checked by Editage (the certificate is attached).
> General comments:
> •Overall writing: the writing style and appropriate usage of the English language can be improved to obtain a higher quality of academic writing and to enable better readability of the manuscript. For example, line 55 –59: “So far, genetic analyses including GWASs revealed schizophrenia-associated sequence variations related to dopamine transmission, such as DRD2 [7], COMT encoding catechol-O-methyltransferase [8], DISC1 encoding disrupted in schizophrenia 1 [9], and PCLO encoding piccolo [10,11,12], whose single nucleotide variations (SNVs) that increase the risk of this disorder have been identified[reviewed in 5 and 13].”or line 104 –106: “In the developmental stage, microglia promote synaptogenesis and eliminate unnecessary synapses (a process called “synaptic pruning”) during different periods of circuit development[37, 38]”. In addition, when using abbreviation, the authors should make sure to write them out at least once in the manuscript (e.g., mEPSP).
Response:
We have revised according to your comment in lines 47–50 and 101–103 to improve readability and avoid misunderstanding. Additionally, we have defined mEPSP as “miniature excitatory postsynaptic potential” in line 171 of the revised manuscript and removed some unnecessary abbreviations.
> •While the authors extensively refer to other studies, they rarely provide details on the applied methods or exact results, i.e., findings from other studies are often formulated in a way that leaves them ambiguous which increases the risk of missing important information. Some examples:
o Line 83 –84: “Cognitive functions related to PFC, including attention, cognitive flexibility, and working memory, are disturbed [29-32], which has been confirmed by functional magnetic resonance imaging [33, 34].” –specify in what way cognitive functions are disturbed and how it has been confirmed by fMRI. Additionally, neither ref. [33] nor ref. [34] made claims about the cognitive functions mentioned by the author –this illustrates the importance on providing details on the exact results.
Response:
According to your suggestion, we have added information regarding the cognitive task and fMRI experiments and phenotype of the patients with schizophrenia to lines 74–80 of the revised manuscript as follows.
Patients with schizophrenia exhibit volume loss in the PFC [27, 28] and disturbance of cognitive functions related to the PFC, including attention, cognitive flexibility, and working memory [29-32]. For example, in the Wisconsin Card Sorting Test, which assesses flexibility of thinking associated with the PFC, a higher frequency of perseverative responses has been observed in patients with schizophrenia [31]. Such functional disturbance in the PFC is consistent with the evidence from functional magnetic resonance imaging studies indicating lower activity of the PFC regarding episodic encoding and retrieval [33] and in response to consummatory pleasure [34].
> o Line 144 –148: “Proteomic analysis of the synaptic microregion revealed that the local expression of proteins related to neuronal transmission, energy metabolism, and the antioxidant system was altered in the C1q-tagged synaptic fraction [70]. Although it is unclear whether such an alteration is a trigger or a consequence of the tagging of C1q, the properties of tagged synapses were demonstrated to differ from those of untagged synapses.” –specify in what way the local expression of these proteins is altered and how the properties of tagged and untagged synapses differ from each other. Additionally, ref. [70] found that out of 410 spots, only 25 exhibited significant differences between tagged and untagged samples, which is not a particularly strong result –this illustrates the importance on providing details on the applied methods.
Response:
We have described the method used to analyze the single synaptosome in line 140 of the revised manuscript. It is true that the number of proteins differed between tagged and untagged synaptosomes. However, we agree with the authors of the study in that the importance of this difference cannot be ignored. We have discussed this in lines 142–144 of the revised manuscript as follows.
Proteomic analysis of the individual synaptosomes conducted using flowcytometry revealed that the local expression of proteins related to neuronal transmission, energy metabolism, and the antioxidant system was altered in the C1q-tagged synaptic fraction [73]. Although it is unclear whether such alteration is a trigger or consequence of C1q tagging, these proteins altered in C1q-tagged synaptic fraction possibly change the neuronal transmission of synapses.
> o Line 284 –285: “Recently, a phase 2, multicenter, randomized, double-blind, placebo-controlled study offered a positive result.” –specify what this ‘positive result’ entailed.
Response:
The outcome of the clinical study for E6011 has been specified in lines 310–315.
Recently, a phase 2, multicenter, randomized, double-blind, placebo-controlled study of E6011 for rheumatoid arthritis was conducted recruiting 190 patients with active rheumatic arthritis (RA) who inadequately responded to methotrexate, the first-choice drug for RA treatment. Administration of E6011 once every two weeks for 24 weeks significantly improved the clinical RA score evaluated using the American College Rheumatology 20% improvement criteria without severe adverse events [146, 147]. Clinical trial of E6011 for Crohn’s disease has also started [148].
In addition, according to your comment, we considered it necessary to further describe the clinical study on pegcetacoplan in lines 302–307 as follows.
However, eculizumab stochastically increases C3 expression in erythrocytes to mediate the formation of MAC leading de novo extravascular hemolysis in patients with PNH, which reduced the clinical benefits of the treatment [143]. To overcome this side effect, pegcetacoplan, a peptide-based C3 inhibitor, was developed and administered in an open-label, phase Ib, prospective, non-randomized study [144]. Pegcetacoplan was confirmed to be well-tolerated and resulted in an improved hematological response in patients with PNH who remained anemic during treatment with eculizumab [144]
> •The main concepts introduced in this manuscript are not clearly linkedto each other and often appear to be a lose enumeration of information. For example: “Psychological stress also increases the risk of SCZ” –the added value of this information is unclear given that this concept is not mentioned again in the manuscript/has not been linked to other concepts such as complement or microglia. It is strongly suggested to provide a clear link between the individual concepts. See,for instance,Germann, M., Brederoo, S. G., & Sommer, I. E. (2021). Abnormal synaptic pruning duringadolescence underlying the development of psychotic disorders.Current opinion in psychiatry.for a nice example on how to link complement, microglia, stress, astrocytes and cognitive deficits in SCZ.
Response:
Thank you for noting that this passage appears to be out of place. The sentence you indicated (“Psychological stress also increases the risk of SCZ” in line 197 of the original manuscript) has been removed following your suggestion. The reference you indicated (“Abnormal synaptic pruning during adolescence underlying the development of psychotic disorders.” by Germann M et al in Curr Opin Psychiatry in 2021) is important for our review, and we have cited it in line 235 as [113] in the revised manuscript.
- Germann M, Brederoo SG, Sommer IEC. Abnormal synaptic pruning during adolescence underlying the development of psychotic disorders. Curr. Opin. Psychiatry 2021, 34, 222-227, doi: 10.1097/YCO.0000000000000696.
> •Although the reference list of this manuscript is quite extensive, more than 60% references are from 2016 or older. Taking into account the main aim of this manuscript to “offer recent knowledge of SCZ [...]”, it is suggested to substitute older references with more recent literature to meet the aim of the manuscript. For example:
> o Reference [1] on the global prevalence of SCZ stems from 2005.
Response:
We have replaced the outdated epidemiological reference with recent ones. In the revised manuscript, references [1, 4, 8, 24, 37, 40, 44, 70, 88-90, 113, 114, 131-140, 147, 148, 151, 152] have been added or replaced to update the information that supports our main arguments; 19 of the 27 newly cited references have been published in or after 2017.
> o The presented view on the microglial pro-inflammatory phenotype is outdated(see for instance Dubbelaar ML, KrachtL, Eggen BJL, Boddeke EWGM. The kaleidoscope of microglial phenotypes. Front Immunol 2018; 9:1753. doi: 10.3389/fimmu.2018.01753.).
Response:
The description of the microglial pro-inflammatory phenotype has been revised as shown in lines 97–99 based on reference (“The Kaleidoscope of Microglial Phenotypes.” by Dubbelaar,. et al in Front. Immunol. in 2018) as follows.
Microglia are activated in disease- and aging-specific manners to release inflammatory cytokines and chemokines, and remove the deposits of pathologic proteins through phagocytosis [44].
- Dubbelaar, M. L.; Kracht, L.; Eggen, B. J. L.; Boddeke, EWGM. The Kaleidoscope of Microglial Phenotypes. Front. Immunol. 2018, 9, 1753, doi: 10.3389/fimmu.2018.01753.
> •This manuscript could benefit from summarizing Beth Stevens line of research on complement mediated microglial synaptic pruning.
Response:
We appreciate your suggestion to incorporate this beneficial line of research and have added reference (“Neuron-Glia Signaling in Synapse Elimination.” by Wilton, D. K. et al in Annu. Rev. Neurosci. in 2019) to line 85 as [40] in the revised manuscript.
- Wilton, D. K.; Dissing-Olesen, L.; Stevens, B. Neuron-Glia Signaling in Synapse Elimination. Annu. Rev. Neurosci. 2019, 42, 107-127, doi: 10.1146/annurev-neuro-070918-050306.
> Introduction:
- This manuscript clearly supports the excessive pruning hypothesis on SCZ as proposed by Feinberg. In the introduction, however, the dopamine hypothesis on SCZ is discussed. The added value of this discussion is unclear, especially given that this topic does not come back later on in the manuscript. Given the main topic of the manuscript “synapse-microglia interaction” and the hypothesis stating that excessive synaptic pruning mediated by microglial overactivation play a role in the development of SCZ, it would be more relevant to mention that current antipsychotic medication is not effective in treating cognitive symptoms (after all, excessive pruning is expected to underlie the cognitive functioning deficits in SCZ).
Response:
We have revised and minimized the description of the dopamine hypothesis as shown in the second paragraph of the Introduction in line with your suggestion. In addition, we have added the following sentences to lines 84–86 and line 89 to avoid confusing the readers.
In a recent decade, synaptic pruning by microglia has been revealed to be involved in psychiatric disease in addition to normal brain development [38, 39, 40].
In the following sections, we introduce microglial functions in synaptic development and maintenance focusing on synaptic pruning and their relevance to schizophrenia pathology.
> •The conclusions drawn by the authors at the end of the introduction (i.e., “Thus, Feinberg’s suggestion is supported by recent immunological findings in schizophrenia.”) is not properly backed up; the paragraph does not contain a single immunological finding to base this conclusion on.
Response:
We have added information regarding immune activation in the PFC of patients with schizophrenia to lines 83–84 based on reference (“Increased inflammatory markers identified in the dorsolateral prefrontal cortex of individuals with schizophrenia” by Fillman, S. G. et al in Mol. Psychiatry in 2013) as [37] following your suggestion.
Furthermore, microglial activation is observed in the PFC of patients with schizophrenia [37].
- Fillman, S. G.; Cloonan, N.; Catts, V. S.; Miller, L. C.; Wong, J.; McCrossin, T.; Cairns, M.; Weickert, C. S. Increased inflammatory markers identified in the dorsolateral prefrontal cortex of individuals with schizophrenia. Mol. Psychiatry 2013, 18, 206-214, doi: 10.1038/mp.2012.110.
> •The statements made by the authors in lines 75 –77 are not backed up properly with references.
Response:
We added reference (“In schizophrenia, immune-inflammatory pathways are strongly associated with depressive and anxiety symptoms, which are part of a latent trait which comprises neurocognitive impairments and schizophrenia symptoms” by Al mulla, et al in J. Affect. Disord. in 2021) as reference [24] for background (line 68).
- Almulla, AF.; Al-Rawi, KF.; Maes, M.; Al-Hakeim, HK. In schizophrenia, immune-inflammatory pathways are strongly associated with depressive and anxiety symptoms, which are part of a latent trait which comprises neurocognitive impairments and schizophrenia symptoms. J. Affect. Disord. 2021, 287, 316-326, doi: 10.1016/j.jad.2021.03.062.
> Potential diagnosis/medication for schizophrenia based on microglia-synapse interaction:
- The authors provide a well-structured and clear description of perspectives for diagnosis of SCZ. However, the section on perspective for medication drug therapy in SCZ needs to be improved:
oThe authors mention that eculizumab, a drug for paroxysmal nocturnal hemoglobinuria (PNH), has been approved by the FDA. Is there any literature on the exact effect of eculizumab in SCZ? What is the (expected)beneficial effect on behavioral/cognitive functioning in SCZ? oThe authors further discuss E6011 which has been developed for rheumatic diseases and mention that a phase 2, multicenter, randomized, double-blind, placebo-controlled study showed a positive result. What is this ‘positive result’(beneficial to mention effect sizes or similar)? In which population(s)was this drug tested? What is the (expected) beneficial effect on behavioral/cognitive functioning in SCZ?
Response:
We have modified lines 315–318 and have clarified that the clinical trials of these drugs targeting schizophrenia have not begun. Therefore, E6011 has not been approved by the FDA even for rheumatoid arthritis. We appreciate your comment regarding our overstatement for the clinical studies.
Although clinical studies have not been initiated for schizophrenia, modulators of the C3-CR3 and CX3CL1-CX3CR1 pathways could be candidates for future medications for the behavioral and cognitive symptoms of schizophrenia mediated by decelerating excessive synaptic pruning.
o The authors proposed CD47 as a stimulant for the “don’t-eat-me” signal which is predicted to decrease synaptic pruning. Are there any studies which investigated this drug (possibly in SCZ)? If so, what were the outcome(s)? Furthermore, if the assumption is that excessive synaptic pruning in SCZ occurs during adolescence/young adulthood, this drug is rather utilized as preventive tool, not as a treatment for SCZ.
Response:
Currently, there is no report of clinical drugs that stimulate the CD47 signal. To express the potential of CD47-targeting medicine, we have added references pertaining to rodent studies to lines 319–323. As you mentioned, modulation of synaptic pruning takes place before the emergence of clinical symptoms. Thus, intervention should be performed in the pre-clinical stage based on our notion. We have discussed this in lines 324–328.
The activation of the don’t-eat-me signal should also be effective for schizophrenia medication. The stimulation of this signal by CD47-SIRPa mediates escape from synaptic pruning [149, 150]. In preclinical studies, deficiency of CD47 induced impairment of social behavior [151] and exaggerated cognitive dysfunction in mice [152]. Agonism of SIRPa could be a potential strategy to suppress synaptic pathology in schizophrenia.
Considering that excessive synaptic pruning occurs before the emergence of clinical symptoms in schizophrenia, the strategy to modulate synaptic pruning should be started preclinically. Thus, such medication should be combined with the monitoring of the synaptic status in the brain as above mentioned. In addition, there is an increased risk of infection because of the suppression of innate immunity, which could ensue with these medication candidates.
o Both minocycline and N-acetyl-cysteine (NAC) have been demonstrated to be effective, safe, and well-tolerated in SCZwith few side-effects. Why havethese drugs not been mentioned?
Response:
We appreciate your beneficial suggestion. Discussion on minocycline and NAC has been added to lines 282–295 as follows.
Clinical trials of anti-inflammatory drugs for schizophrenia have been performed. Minocycline, a tetracycline-type antibiotic, suppresses the microglial inflammatory response [131, 132]. Minocycline is reported to rescued cognitive dysfunction and social behavioral impairment in animal models of schizophrenia [133, 134]. Minocycline abolished the phagocytosis of patient-derived iMG toward spine density on neurons differentiated by patient-derived iPSCs [88]. In clinical trials, minocycline treatment improved the working memory [135] and verbal and visual learning [136] of patients with schizophrenia. Although minocycline is infrequently prescribed partly because of the increased risk of autoimmune disease [137], these outcomes demonstrate the clinical effectiveness of this strategy targeting microglia in schizophrenia. N-acetylcysteine (NAC), a precursor of antioxidant glutathione, exerts a wide range of protective effects, such as the regulation of oxidative status, inflammation, and monoamine neurotransmission in rodent models and human patients [138, 139]. NAC showed a beneficial effect mainly on the negative symptoms of patients with schizophrenia receiving antipsychotic treatment (comprehensively reviewed in [140]). These therapeutic effects of NAC are possibly achieved by multiple mechanisms, such as modulation of the oxidative and inflammation statuses and neurotransmission.
- Miyanohara, J.; Kakae, M.; Nagayasu, K.; Nakagawa, T.; Mori, Y.; Arai, K.; Shirakawa, H.; Kaneko, S. TRPM2 Channel Aggravates CNS Inflammation and Cognitive Impairment via Activation of Microglia in Chronic Cerebral Hypoperfusion. J. Neurosci. 2018, 38, 3520-3533, doi: 10.1523/JNEUROSCI.2451-17.2018.
- Shin, D. A.; Kim, T. U.; Chang, M. C. Minocycline for Controlling Neuropathic Pain: A Systematic Narrative Review of Studies in Humans. J. Pain Res. 2021, 14, 139-145, doi: 10.2147/JPR.S292824.
- Mizoguchi, H,; Takuma, K.; Fukakusa, A.; Ito, Y.; Nakatani, A.; Ibi, D.; Kim, H. C.; Yamada, K. Improvement by minocycline of methamphetamine-induced impairment of recognition memory in mice. Psychopharmacology 2008, 196, 233-241, doi: 10.1007/s00213-007-0955-0.
- Zhu, F.; Liu, Y.; Zhao, J.; Zheng, Y. Minocycline alleviates behavioral deficits and inhibits microglial activation induced by intrahippocampal administration of Granulocyte-Macrophage Colony-Stimulating Factor in adult rats. Neuroscience 2014, 266, 275-281, doi: 10.1016/j.neuroscience.2014.01.021.
- Kelly, D. L.; Sullivan, K. M.; McEvoy, J. P.; McMahon, R. P.; Wehring, H. J., Gold, J. M.; Liu, F.; Warfel, D.; Vyas, G.; Richardson, C. M.; et al. Adjunctive Minocycline in Clozapine-Treated Schizophrenia Patients With Persistent Symptoms. J. Clin. Psychopharmacol. 2015, 35, 374-381, doi: 10.1097/JCP.0000000000000345.
- Zhang, L. L.; Zheng, H. B.; Wu, R. R.; Kosten, T. R.; Zhang, X. Y.; Zhao, J. P. The effect of minocycline on amelioration of cognitive deficits and pro-inflammatory cytokines levels in patients with schizophrenia. Schizophr. Res. 2019, 212, 92-98, doi: 10.1016/j.schres.2019.08.005.
- Dominic, M. R. Adverse Reactions Induced by Minocycline: A Review of Literature. Curr. Drug Saf. 2021, doi: 10.2174/1574886316666210120090446.
- Fan, C. Q.; Long, Y. F.; Wang, L. Y.; Liu, X. H.; Liu, Z. C.; Lan, T.; Li, Y.; Yu, S. Y. N-Acetylcysteine Rescues Hippocampal Oxidative Stress-Induced Neuronal Injury via Suppression of p38/JNK Signaling in Depressed Rats. Front. Cell. Neurosci. 2020, 14, 554613, doi: 10.3389/fncel.2020.554613.
- Steullet, P.; Cabungcal, J. H.; Monin, A.; Dwir, D.; O'Donnell, P.; Cuenod, M.; Do, K. Q. Redox dysregulation, neuroinflammation, and NMDA receptor hypofunction: A "central hub" in schizophrenia pathophysiology? Schizophr. Res. 2016, 176, 41-51, doi: 10.1016/j.schres.2014.06.021.
- Smaga, I.; Frankowska, M.; Filip, M. N-acetylcysteine as a new prominent approach for treating psychiatric disorders. Br. J. Pharmacol. 2021, doi: 10.1111/bph.15456.
Conclusion:
- This manuscript aims to provide new insights regarding diagnosis and medication for SCZ based on neuronal synapse-microglia interaction.The conclusion, however, is formulated in a way that leaves insights for diagnosis and medication for SCZ ambiguous and vague.
Response:
We have modified the Conclusion section to avoid ambiguity as shown in lines 336–344 following your suggestion.
Various molecules, such as CX3CL1-CX3CR1, CD47-SIRPa, and lectins, are physiologically and pathologically involved in this process. Recent publications indicating the contribution of complement components to schizophrenia are attracting attention regarding this relationship. Monitoring and modulation of microglial synaptic pruning based on the complement system would be powerful tools for diagnosis and medication in schizophrenia. Further investigation on synapse–microglia interaction could reveal other molecular targets for diagnosis and medication. Considering that schizophrenia is a polygenic disease, such medication and diagnosis require to be combined to find excessive pruning in the preclinical stage, and early medication could maximize the therapeutic effects for schizophrenia.
- In the introduction of this manuscript,it is stated that “In addition, a number of biomarker studies measuring inflammatory cytokines in the peripheral blood and cerebrospinal fluid (CSF) [21, 22] and imaging studies using positron emission tomography (PET) to detect neuroinflammation [23, 24] have clarified the immunological propertiesof patients with schizophrenia.”–this is in conflict with the last sentence of the conclusion statingthat an immunological understanding has not yet been reached.
Response:
We appreciate you comment regarding this contradiction. The last sentence of the Conclusion has been revised as follows (lines 343–344).
Development of the medication and diagnosis focusing on synapse–microglia interaction and its clinical application to schizophrenia are desired.
